# *Cucumber mosaic virus* Is Unable to Self-Assemble in Tobacco Plants When Transmitted by Seed

**DOI:** 10.3390/plants11233217

**Published:** 2022-11-24

**Authors:** Antonella Vitti, Israel Pagán, Brigida Bochicchio, Angelo De Stradis, Pasquale Piazzolla, Antonio Scopa, Maria Nuzzaci

**Affiliations:** 1School of Agricultural, Forestry, Food and Environmental Sciences, University of Basilicata, Viale dell’Ateneo Lucano 10, 85100 Potenza, Italy; 2Department of Pharmacy, University of Salerno, Via Giovanni Paolo II, 132, Fisciano, 84084 Salerno, Italy; 3Centre for Plant Biotechnology and Genomics UPM-INIA/CSIC, Polytechnic University of Madrid, Campus Montegancedo, M40 Highway km.38, Pozuelo de Alarcón, 28223 Madrid, Spain; 4Laboratory of Protein Chemistry, Laboratory of Bioinspired Materials (LaBIM), Department of Science, University of Basilicata, Viale dell’Ateneo Lucano, 10, 85100 Potenza, Italy; 5Institute for Sustainable Plant Protection, CNR, Via Amendola 122/D, 70126 Bari, Italy

**Keywords:** virus vertical transmission, CMV-Fny strain, pseudorecombinant virus, chimeric virus, infection rate, seed-growth tests, electron microscopy, circular dichroism spectroscopy, viral assembly

## Abstract

*Cucumber mosaic virus* (CMV), which has great impact on agronomic production worldwide, is both aphid and seed transmitted. Although the mechanisms of aphid transmission have been widely studied, those underlying the ability of CMV to survive and remain infectious during the passage from one generation to the next through the seeds are still to be clarified. Moreover, the viral determinants of seed transmission rate are poorly understood. Three viral genotypes produced from same RNA 1 and 2 components of CMV-Fny but differing in RNA 3 (the wild type CMV-Fny, a pseudorecombinant CMV-Fny/CMV-S and a chimeric CMV previously obtained by our group, named F, FS and CS, respectively) were propagated in *Nicotiana tabacum* cv Xanthi plants in order to assess differences in tobacco seed transmission rate and persistence through plant generations in the absence of aphid transmission. Seed-growth tests revealed CMV infection in the embryos, but not in the integuments. Seedlings from seed-growth tests showed the presence of all considered viruses but at different rates: from 4% (F, FS) to 16% (CS). Electron microscopy revealed absence (CS) of viral particles or virions without the typical central hole (F and FS). In agreement, structural characteristics of purified CMV particles, assessed by circular dichroism spectroscopy, showed anomalous spectra of nucleic acids rather than the expected nucleoproteins. These alterations resulted in no seed transmission beyond the first plant generation. Altogether, the results show for the first time that correct virion assembly is needed for seed infection from the mother plant but not to seedling invasion from the seed. We propose that incorrect virion formation, self-assembly and architecture stability might be explained if during the first stages of germination and seedling development some tobacco seed factors target viral regions responsible for protein-RNA interactions.

## 1. Introduction

*Cucumber mosaic virus* (CMV, family *Bromoviridae*, genus *Cucumovirus*) is a plant virus with great impact on agronomic production worldwide due to its extremely wide host range and geographical distribution [1]. CMV is a positive-sense RNA virus whose genome is encapsidated in icosahedral particles of about 30 nm in diameter. The virus genome consists of three single-stranded RNAs (RNA 1, 2 and 3), all necessary for infectivity. RNA 1 encodes for the 1a protein, involved in virus replication together with the 2a protein, which is encoded by RNA 2. RNA 2 also harbors the 2b gene, which encodes for a protein involved in suppressing the host RNA silencing defense response. RNA3 encodes for the movement protein (MP) and the coat protein (CP), the latter being expressed through the subgenomic RNA 4. CMV is mainly transmitted horizontally from plant-to-plant by aphids, but vertical transmission from-parent-to-offspring through seeds was reported few years after the virus was first described in cucumber [2,3,4]. In the last twenty years, CMV vertical transmission has been described in several plant species, such as some legumes [5,6,7,8,9], spinach (*Spinacia oleracea* L.) [10], pepper (*Capsicum annuum* L. cv Marengo) [11] and *Arabidopsis thaliana* [12,13,14]. Thus, seed transmission is an essential mode for CMV persistence (the virus can survive within the seed as long as this remains viable) and long-distance dissemination (even at trans-continental scale). Moreover, it also provides an inoculum source for subsequent spread by aphids, having substantial epidemiological effects and therefore a far-reaching impact on crop production [10,15]. Indeed, it has been shown that very low CMV seed transmission rates are enough to start an outbreak [15].

In *A. thaliana*, it has been determined that the efficiency of vertical transmission depends on the host-virus genotypes per genotype interaction [14]. In agreement, another recent study conducted on *Capsicum* species reported that the CMV ability to be transmitted by seeds, and the rate of seed transmission, are cultivar-dependent [16]. The host determinants controlling CMV seed transmission are not well understood yet. In *A. thaliana*, virus multiplication and speed of within-host movement through the inflorescence have been shown to chiefly predict the CMV seed transmission rate [14]. In addition, Genome-Wide Association Studies (GWAS) indicated that the same host functions that modulate CMV virulence (i.e., the effect of pathogen infection on host progeny production) are important for CMV seed transmission rate [17]. However, the host genes controlling virus multiplication and movement have not been identified, and GWAS results need to be experimentally validated. On the virus side, as soon as 1992, a study with pseudorecombinants obtained from strains that were, or were not, seed transmissible in *Phaseolus vulgaris* demonstrated that RNA 1, co-governing viral replication and affecting virus movement, has a great influence on CMV transmissibility by seeds [5]. These results are compatible with those of works focused on the plant side. Therefore, RNA 1 is considered to affect the efficiency of seed transmission. However, Pagán (2019) [15] suggested that RNAs 2 and 3 could also represent viral genetic determinants of CMV seed transmission, although there is no experimental test of this hypothesis, and information on the amino acids potentially involved in this process is not available. Efforts have been made to explore whether the virus is vertically transmitted over several generations [13]. However, in line with the lack of information on the genetic determinants of CMV seed transmission, the mechanisms underlying the ability of CMV to survive and remain infectious in seeds across plant generations have not been defined. 

This work aims to better understand the role played by the proteins encoded by the genomic segments RNA 1, 2, and 3 (and particularly the latter segment) in CMV seed transmission, by searching for possible unknown virus determinants, as well as by clarifying if the virus has the capacity to remain viable across host generations and what are the mechanisms underlaying this ability. To address these questions, we used *Nicotiana tabacum* as a model host and three different virus genotypes: CMV-Fny (F), a pseudorecombinant CMV-Fny/CMV-S (FS) and a chimeric CMV previously obtained by our group (CS) [18]. All genotypes have RNA 1 and 2 components from CMV-Fny, and different RNA 3: of CMV-Fny (F), of CMV-S (FS) and a version of the CMV-S mutated in the amino acid 131 of CP and carrying an exogenous peptide of 11 aa (CMV_392_, CS). Using these genetic materials, we analyzed the location of CMV within the seed, we explored the relationship between modifications in RNA 3 and seed transmission rate and we determined for how many generations CMV persisted in the plant through strict vertical transmission.

## 2. Results

### 2.1. Symptoms of Virus Infection in Mother Plants

Fourteen days after inoculation, as well as during seed harvesting, plants inoculated with all viral genotypes showed typical infection symptoms (Figure 1a,b,d,e,g,h) compatible with those reported previously [18], thus proving that the viruses replicated, assembled and systemically spread. Virus presence was corroborated by molecular and serological means (see Section 2.3). Seeds obtained from tobacco mother plants infected with the wild type CMV-Fny (F), the pseudorecombinant CMV-Fny/CMV-S (FS) and the chimeric CMV_392_ (CS) were collected 120–135 days post inoculation (d.p.i.). All seedlings obtained by the germination of these seeds resulted symptomless (Figure 1c,f,i).

As a control, virions from the purified three CMV genotypes, as well as systemically infected tissues of mother plants were used to mechanically inoculate healthy plants. In all cases, RT-PCR and serological analyses revealed the presence of the replicase gene and CP in all infected plants 14 d.p.i. The typical CMV symptoms were also shown and virions were observed under EM (data not shown).

### 2.2. Location of CMV within Infected Tobacco Seeds

Grow-out tests of first-generation seeds allowed us to separate seed integuments from embryos and then to detect the presence of CMV infection in the two different tissues (Figure 2). 

Both RT-PCR and Immuno-dot blot (IB) analyses revealed seed infection only in the embryo but not at the integument level, as shown in Figure 2. Specifically, RT-PCR analysis of the separated seed tissues showed the presence of amplicons of the expected size (513 bp) for all nucleic acids extracted from plants infected by the three viral genotypes F, FS and CS, so confirming the presence of the RNA-dependent RNA polymerase (*RdRp*) gene only in the embryos and never in the integuments (Figure 2a). Similarly, IB analysis confirmed this finding through detecting the viral CP by CMV polyclonal antiserum (Figure 2b).

### 2.3. CMV Seed Transmission Rate in Tobacco

First-generation seedlings, obtained from the germination in pot of seeds derived from CMV-infected mother plants, were monitored for virus presence after 6–8 weeks post-germination, and were used to analyze CMV seed transmission rate in tobacco plants. For comparison purposes, virus infection in mother plants was also analyzed.

For all three considered viruses, the analyses on mother plants during seed collection and on first-generation seedling leaf tissues showed the presence of both *RdRp* gene by RT-PCR (Figure 3a) and CP by IB (Figure 3b).

For each considered viral genotype, 400 seedlings derived from mature seeds collected from the 30 infected mother plants were randomly chosen and analyzed by IB, in order to establish the CMV vertical transmission rate. The results indicated that the CMV seed-transmission rate ranged from 4% for F and FS to 16% in the case of CS, in spite of symptomless infection in all seedlings (Figure 1c,f,i).

Purified virions from both mother plants and first-generation seedlings were analyzed by electron microscopy (EM), as reported in Figure 4. In mother plants, all CMV viral genotypes displayed the characteristic lesser density central region, named “hole” (Figure 4a). On the contrary, when virions derived from first-generation seedlings were analyzed, they were absent (CS) or the viral particles had not the typical central hole (F and FS) (Figure 4b).

Figure 5 shows circular dichroism (CD) spectra of purified virions isolated from mother plants and first-generation tobacco seedlings. The conformational features are those expected for the typical CMV bands when obtained from mother plants (Figure 5a). CD spectra of viruses extracted from mother plants showed a strong positive band at about 200 nm, two negative bands at about 208 nm and 220 nm, and a small positive band at about 270 nm. Both the positive, at 200 nm, and negative band, at 208 nm, reached the maximum ellipticity value in the case of F, and the minimum for CS, with FS showing intermediate values (Figure 5a). In contrast, the positive band at about 270 nm displayed the same intensity for all three viruses. CD spectra of first-generation seedling viruses are shown in Figure 5b. CMV viral RNA was used to compare the CD spectra of these viruses. All CD spectra were similar to the RNA spectrum, with some exceptions for CS. They were characterized by a positive peak below 200 nm even if with lower ellipticity values than those shown in Figure 5a. A positive band at about 270 nm, with the highest ellipticity for F and values around zero for FS, was found (Figure 5b). Furthermore, two negative bands at about 240 and 200 nm were evident for all samples, except CS. The RNA CD spectrum shows a trend toward positive values at about 220 nm, analogously to F and FS that showed positive values and a red wavelength shift. Summarizing, a typical CD profile of a CMV nucleic acid, rather than the expected nucleoprotein, was revealed for all three virus genotypes in the first-generation seedlings (Figure 5b).

### 2.4. Second Generation Seeds and Seedlings

Second-generation seedlings, obtained from the germination in pot of seeds deriving from 30 CMV-infected first-generation plants, were monitored after 6–8 weeks post germination. In all 400 s-generation seedlings, none of the CMV genotypes were detected for either by RT-PCR or by IB analyses. Similarly, purified F, FS and CS particles were never observed by ME. These findings indicate that, in tobacco, the vertical transmission of CMV was unable to proceed beyond the first generation.

## 3. Discussion

*Cucumber mosaic virus* can be transmitted both horizontally and vertically, by aphids and seeds, respectively. To develop efficient strategies to control virus outbreaks, understanding the mechanisms and determinants underlying the ability of CMV to survive and remain infectious in seeds and to persist during strict vertical passage from one generation to the next, are of pivotal importance. However, CMV seed-borne infections and their implications for virus epidemiology, as well as the factors and processes that affect vertical transmission, are not fully understood yet [15].

In the present study, we show that in tobacco the CMV seed-borne infection was associated with the virus presence in the embryo for the wild type, the pseudorecombinant and the chimeric CMVs (Figure 2). This is probably due to virion instability, which reduces viability outside living cells and therefore on seed coats [9], although host-specific factors may also be involved. Indeed, in other plant species the embryo infection is not necessary for seedling infection. For instance, using CMV-Fny as in this work, Ali and Kobayashi (2010) [11] reported a high rate of seed coat infection linked to virus vertical transmission. These authors also reported embryo infection at low rates and showed that embryo infection had not guaranteed CMV seed transmission, which was attributed to an inactivation suffered by the virus during the seedling germination phase [11]. Regardless the route of transmission, our findings seem to confirm that seed invasion and virus multiplication into the seed are necessary (but not sufficient) for CMV transmission, which would be therefore affected by RNA 1 and 2, through their participation in viral replication and movement from mother plants to seeds (Figure 2 and Figure 3) [5]. 

Results regarding the seedling infection rate suggest the existence of potential viral genetic determinants of seed transmission in CMV RNA 3. Specifically, seed-borne infection transferred to seedlings ranged from 4% for F and FS to 16% in the case of CS. These values overlap with those reported by other researchers on CMV in several plant species, such as incarnate clover and pumpkin (5%) [8,19] or the above-mentioned pepper (from 10 to 14%) [11] and spinach (nearly 15%) [10]. Notably, the rate registered in CS quadrupled that of F and FS. As indicated in Section 4.1., all three CMV genotypes were produced starting from the same RNA 1 and RNA 2 components of CMV-Fny; whereas the RNA 3 component was derived from CMV-Fny in F and from CMV-S in FS and CS. This latter chimeric CMV differs from FS by the presence of an exogenous sequence in position 392 of the CP gene, obtained via the site-directed mutagenesis T_391_→G and C_392_→G in the codon for the aa 131 (changed from Ser_131_ to Gly_131_), and located in the βE-αEF region, without affecting to any significant degree the stability of chimeric subunits during serial mechanical plant-to-plant horizontal passages [18]. Therefore, we may point to RNA 3 (and particularly the CP) as a possible viral genetic determinant of CMV seed transmission rate. Moreover, we can also speculate, for the first time, on the eventual amino acids implicated, such as 131 and/or those belonging to βE-αEF region of the CP, provided that the other factors potentially to be considered (i.e., host, time of plant infections, absence of mixed infections with other viruses) are the same for all the studied CMVs and therefore not involved [15]. The possible role of the βE-αEF region in seed transmission rate is also supported by the recognized function of the amino acid 129 of the CP, the first of the βE-αEF region. In fact, this amino acid is considered a virulence determinant and key regulator of symptoms induced in plants by CMV [20,21], with effects that are not host specific, but dependent on the CMV strain [22]. In agreement, in our experiments no symptoms on seedlings derived from infected seeds by all virus genotypes were observed (Figure 1c,f,i), as already demonstrated for other species in which CMV is seed transmitted [11,23]. Nevertheless, the absence of symptoms observed in first-generation seedlings in this work may well be related to an involvement of βE-αEF region of the CP due to incorrect or lack of virion assembly, as observed for all three genotypes F, FS and CS by EM and CD spectra (Figure 4b and Figure 5b). The βE-αEF region of the CP is a flexible loop and this property is conferred by its first amino acid 129 [24,25]. Suzuki et al. (1995) [26] observed that the substitution of the amino acid 129 from Serine to Phenylalanine (S_129_ → F_129_) in the CP of the CMV-Y strain is able to disrupt the virion assembly and provoke the aggregation of CP molecules, to elicit necrosis in tobacco plants. In that amino acid substitution, a polar uncharged group (S) was replaced with a non-polar uncharged group (F). In the current study, we used a chimeric CMV where a similar substitution occurred in the same βE-αEF region of the CP, but in the nearby amino acid position 131. In fact, in the genotype CS, the amino acid Serine at position 131 of the CP was replaced with a non-polar uncharged group, that is the aa Glycine (S_131_→G_131_). Besides the mentioned substitution, and starting from the same position, an exogenous peptide of 11 amino acids was also inserted in the CP. These modifications resulted in some variation of the electrostatic potential in the CP, which is known to play an important role in the protein folding and stability, as well as in the protein-protein and protein-nucleic acid interactions [18,27]. Therefore, these modifications, although they did not interfere with virus stability in serial plant-to-plant passages, as reported in a previous research of this group [18], could be the reason for the total inability to assemble by CS (Figure 4b) under strict vertical transmission. Other studies with mutation of the aa 131 (without carrying the exogenous peptide) or in different positions of the CP (i.e., from 129 to 136, that are the first and the last aa of the βE-αEF region) will be useful for validating the role of the CP in CMV vertical transmission. 

In *A. thaliana*, an autogamous species like the tobacco used in this work, seed transmission rates for three different CMV strains increased after five serial passages of strict vertical transmission, in association with an analogous reduction of virus accumulation and virulence [13,28]. These observations were explained by the authors as a result of a reciprocal co-evolutionary selection/interaction between host and virus during the seed transmission, aimed to reduce the damage in the plant induced by the virus infection, to favor CMV maintenance in the plant population via seed infection, and also to increase plant fitness [13]. Although symptoms were never observed in seedlings of the first generation, our results disagree with a co-evolution towards lower virulence because the virions are not anymore infectious after the first generation, regardless of the differences on seed transmission rate between F and FS and CS in this first progeny. However, our results are in accordance with a study on pea plants contaminated by *Pea seed-borne mosaic virus* (PSbMV) through seed transmission, where infected plants of the second generation expressed no symptom and, at the same time, the virus was not detectable in their vegetative parts [29]. We might think that an extinction of the virus population could have occurred, maybe due to severe population bottlenecks responsible to hamper the virus ability to pass on through generations and preventing the adaptation of all virus genotypes here studied to seed transmission in tobacco. Such bottlenecks during seed transmission have been previously reported [30]. Supporting this idea, we detected few defective virions passing to the first generation, which were unable to recover the virus population and therefore the virus did not invade the seeds again. However, this might not be the only force at play: while F and FS were at least able to form viral particles, although without the typical central hole, CS showed a complete inability to self-assemble (Figure 4b). Notably, both the replicase gene and the CP were detected in seedlings infected with all three CMVs (Figure 3) and the highest seed transmission rate was observed for CS in the first-generation seedlings. This suggests that the capacity of the virus to invade seedlings from infected seeds does not require virion formation and perhaps nude RNA molecules are enough. In contrast, seed invasion from mother plants and/or virus survival within the seed would need correct virion assembly. Indeed, virions of the three CMV genotypes purified from mother plants were infectious and assembled correctly (see Results).

EM analysis and CD spectra indicated analogous peculiarities for the three CMVs in mother plants. For example, the positive band revealed by CD spectra for the purified virions at about 270 nm displayed the same intensity for all three viruses, indicating the same stacking of RNA bases and structural organization [27]. At odds with this, EM and CD analyses on first-generation seedlings confirmed that abnormal virus particles (F and FS) or no virions at all (CS) were detected (Figure 4b and Figure 5b). As above mentioned, the purified virions appeared to be dissociated as a mixture of RNA and proteins interacting with each other, although in a different manner for CS than for the other two CMV genotypes (Figure 5b). Altogether, our findings suggest that for all three viruses, something seems to happen certainly not in the mother plants, but during seed germination and first-generation seedlings development. We could also speculate that some tobacco seed substances able to interfere with some viral components responsible for protein-RNA interactions could be involved. Chemical analysis of tobacco seeds has indicated a high tocopherol content in different genotypes of *Nicotiana tabacum* (L.) [31,32]. These compounds, also known as vitamin E, are considered lipophilic antioxidants with a possible role in activating both basal and induced resistance in plants [33], thanks to the ability to remove reactive oxygen species (ROS) or polyunsaturated fatty acid radical species [34]. In our case, it can be argued that CMV particles are able to infect seeds but, during germination, some substance (e.g., tocopherols contained in the seed) could be responsible for targeting CMV components, such as CP N-terminal region, rich in basic amino acids; this could disturb (F and FS) or disrupt (CS) the correct virion formation, self-assembling and architecture stability in the developed seedlings of first generation [27]. As a consequence, a second round of seed infection turns out to be not possible. Currently, no data are available from sequencing of viruses from first-generation seedlings to establish also a possible shift in the RNAs sequence of the infecting CMV viruses. Therefore, a next step in this research would be focused on checking the integrity of individual virus RNAs and verifying possible mutations responsible for hampering the ability of the virus to pass on through generations during vertical transmission.

## 4. Materials and Methods

### 4.1. Virus and RNA Sources

The wild type CMV-Fny strain (F), pseudorecombinant CMV-Fny/CMV-S (FS) and chimeric CMV_392_ (CS) were used [18]. They were propagated in *Nicotiana tabacum* cv Xanthi plants and virions were purified as described by Lot et al. (1972) [35]. All three CMV genotypes were produced starting from the same RNA 1 and RNA 2 components of CMV-Fny strain; whereas the RNA 3 component was derived from the same Fny strain in F and from CMV-S strain in FS and CS. This latter chimeric CMV differs from FS by the presence of an exogenous sequence in position 392 of the CP gene, obtained via the site-directed mutagenesis T_391_→G and C_392_→G in the codon for the aa 131 (changed from Ser_131_ to Gly_131_) [18]. 

### 4.2. Virus Inoculation and Production of Infected Seeds

Tobacco seeds were sterilized using 1% Na-hypochlorite solution for 1 min and then rinsed with sterile distilled H_2_O, before imbibition on moist filter paper at 4 °C for 24 h in the dark. Seeds germinated on water-dampened filter paper in a sterile Petri dish at 26 °C. One day after germination, seedlings were transferred to sterilized soil-filled pots. Throughout the experiment, plants were kept in a growth chamber with a 16 h photoperiod, at 26/23 °C (day/night), and watered with tap water, according to the needs of the seedlings, daily checked. Ten μg of virions from each purified CMV genotype were used to mechanically inoculate 30 tobacco plants at the four-leaf stage, which were kept in the growth chamber as infected mother plants until obtaining developed and mature tobacco seeds. The mechanically inoculated tobacco plants were monitored for symptom development. Ten uninfected control plants were also kept in a separate growth chamber as mother healthy plants. The seeds were collected as they matured, in a scaled manner from each mother plant, pooled and stored all together. In the same way, 30 infected tobacco plants obtained from first-generation seeds were grown and represented the mother plants for second-generation seeds. Again, second-generation seeds were pooled and stored. During all experimentation, plants were carefully checked to ensure absence of aphids. 

The following samples were tested for virus presence by RT-PCR and Immuno-dot blot (IB), as described in the subsequent subsections: systemically infected tissues of each mother plant (collected at 14 d.p.i. and during seeds harvesting); embryos and seed coats separated after germination of CMV-infected seeds on water dampened filter paper; 400 seedlings of first generation and 400 of second generation obtained from the germination in pot of CMV-infected seeds, 6–8 weeks post germination. Each seedling of first and second generation was derived from seeds randomly chosen from the total sets collected from mother plants and first-generation mother plants, respectively.

CMVs from both tobacco mother plants and first- and second-generation seedlings were purified according to Lot et al. (1972) [35]. All purified CMV particles were analyzed by electron microscopy (EM), and assessed by circular dichroism (CD) spectroscopy.

### 4.3. Reverse Transcription (RT)-PCR Analysis 

Total plant RNAs of samples infected with F, FS and CS were extracted by the PureLink^TM^ Micro-to-Midi Total RNA Purification System (Invitrogen, Milan, Italy). Five hundred ng were reverse transcribed and amplified in a single tube using the SuperScript^TM^ III One-Step RT-PCR System with Platinum^®^
*Taq* DNA Polymerase (Invitrogen). The RT-PCR reaction mixture (a final volume of 50 μL) was prepared as described by the manufacturer adding 1 μL of both the reverse and forward 10 μM primers. The following pairs of primers were used: P_RevRep_ (5′-CCATCACCTTAGCTTCCATGT-3′), complementary to position 1895–1915 of the CMV RNA-dependent RNA polymerase (*RdRp*) gene, and P_ForRep_ (5′-TAACCTCCCAGTTCTCACCGT-3′), homologous to position 1403–1423 of the CMV *RdRp* gene (accession NC_002035), according to Grieco et al. (2000) [36]. The PCR fragments were fractionated on 1.5% agarose gel and stained with SYBR Safe^TM^ DNA gel stain (Invitrogen).

In the case of seedlings obtained from germination of CMV-infected seeds in pots, pairs of leaves for each plantlet (of the 400 total) were harvested, pooled in groups of ten (a total of 40 pool), and mixed for total nucleic acid extraction. When CMV was detected in a pool, total RNA extraction and RT-PCR from each single plant were performed, accordingly. 

### 4.4. Immuno-Dot Blot (IB) Analysis

Samples infected with F, FS and CS were homogenized in 0.2 M Tris, 1.5 M NaCl (TBS 10×) and centrifuged at 10,000 rpm for 10 min at 4 °C. The supernatant was gently dropped (2 µL) onto nitrocellulose membrane (Invitrogen). The membrane was air-dried, blocked for 45 min in 5% milk Tween–TBS and incubated for 1 h with CMV polyclonal antiserum (Bioreba AG, Switzerland) (diluted 1:2000). The membrane was then treated with anti-rabbit (Pierce Biotechnology, Rockford, IL, USA) alkaline phosphatase-conjugated antibodies (Abs) (diluted 1:2000), and the reactivity was detected using the Sigma Fast^TM^ kit (Sigma Chemical Co., St. Louis, MO, USA). All incubations were performed at room temperature; after each incubation the membrane was washed three times with 20 mM Tris, 150 mM NaCl, 0.05% Tween-20 (Tween–TBS 1×).

### 4.5. Electron Microscopy (EM)

Purified virions isolated from tobacco plants were examined via negative staining with 2% aqueous uranyl acetate and immediately processed for electron microscopy assays. Observations were performed with a Philips Morgagni 282D electron microscope at 60 kV.

### 4.6. Circular Dichroism (CD) Spectroscopy

Circular dichroism (CD) spectra of purified virion (0.2 mg/mL) samples in aqueous solution were recorded on a Jasco J600 CD spectropolarimeter at 24 °C using a cell with a 1 mm optical path length. A HAAKE water bath was used to control the temperature. Data are expressed as molar ellipticity [*θ*]M in deg cm^2^ dmol^−1^ [37].

## 5. Conclusions

For the first time, this study demonstrated the ability of three different genotypes of CMV to be seed-borne in tobacco plants, and the inability to proceed beyond the first generation of plants. The three different genotypes varied in seed transmission rate and in the genomic sequence at RNA 3. Thus, our data allows a speculation on this genomic segment as a possible viral genetic determinant of seed transmission, and on the amino acids potentially implicated in this process and their position: codon for Ser_131_→Gly_131_ (nt 391–393), located in the βE-αEF region of the CP.

In addition, we show that in tobacco plants CMV is unable to persist for more than one generation of strict vertical transmission. An explanation of mechanisms determining the lack of second-generation seedlings infection is that defective or few virions passing to the first generation are unable to recover the virus population after severe population bottlenecks associated with seed transmission, so that the virus does not invade the seeds again. We hypothesize that during the first stages of germination and seedling development some tobacco seed compounds (i.e., tocopherols) may target, for example, the viral CP N-terminal region, rich in basic amino acids and responsible for protein-RNA interactions. In this way, the correct virion formation, self-assembling and architecture stability are interfered with and cannot occur. As a consequence, the vertical transmission is restricted to the first generation of plants, without the ability to achieve a second round of progeny infection, regardless of the transmission rate.

## Figures and Tables

**Figure 1 plants-11-03217-f001:**
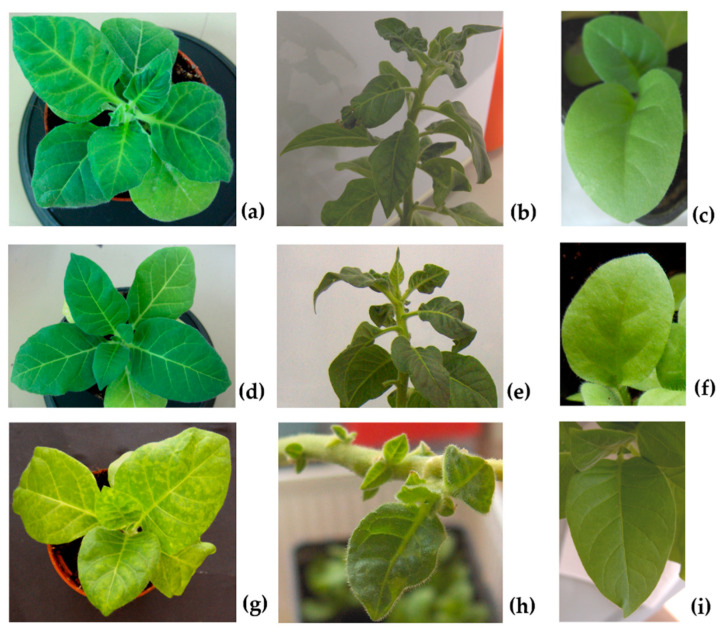
Symptoms induced on *N. tabacum* cv Xanthi mother plants 14 days post-inoculation (first column) and during seeds harvest (second column). Seedlings of the first generation showing no symptoms (third column). Images correspond to plants infected with F (CMV-Fny) (**a**–**c**); FS (CMV-Fny/CMV-S) (**d**–**f**); CS (chimeric CMV_392_) (**g**–**i**).

**Figure 2 plants-11-03217-f002:**
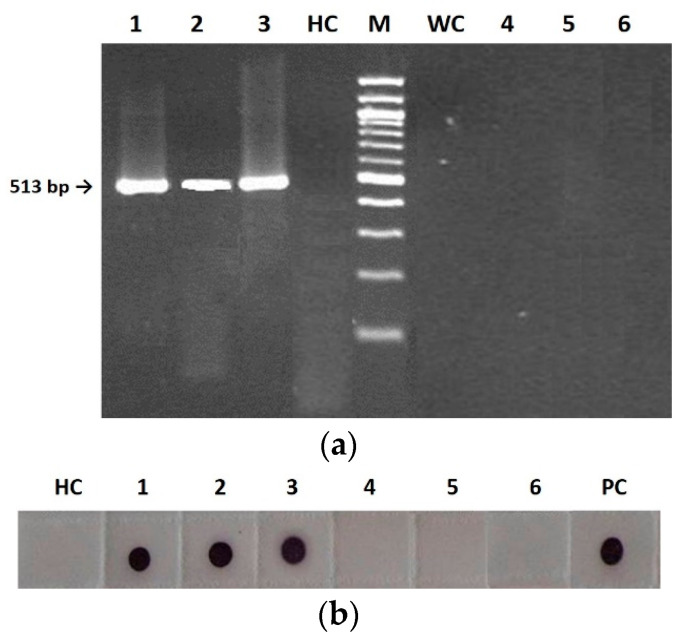
Detection of wild type (F), pseudorecombinant (FS) and chimeric (CS) CMVs in tobacco seed embryonal (1–3) and integumental (4–6) tissues by RT-PCR (**a**) and Immuno-dot blot (**b**) analyses. (**a**): lines 1–3, DNA fragment of 513 bp derived from embryonal tissues infected with F (1), FS (2), and CS (3); HC, healthy control; M, 100 bp DNA Ladder (BioLabs); WC, water control; lines 4–6, no fragment derived from integumental tissues infected with F (4), FS (5), and CS (6). (**b**): membrane probed with CMV polyclonal antiserum (Bioreba AG). HC, healthy control; 1–3, CP revealed from embryonal tissues infected with F (1), FS (2), and CS (3); 4–6, no CP revealed from integumental tissues infected with F (4), FS (5), and CS (6). PC, positive control.

**Figure 3 plants-11-03217-f003:**
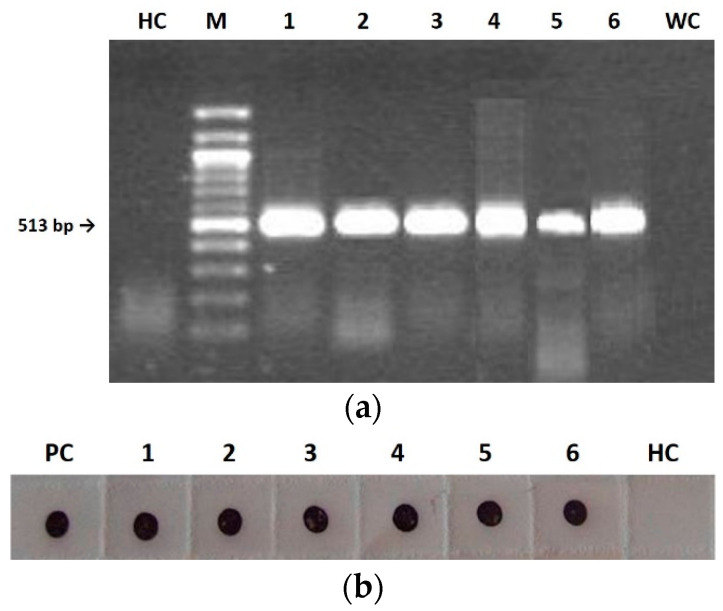
Detection of wild type (F), pseudorecombinant (FS) and chimeric (CS) CMVs in tobacco mother plants (1–3) and first-generation seedlings (4–6) by RT-PCR (**a**) and Immuno-dot blot (**b**). (**a**): HC, healthy control; M, 100 bp DNA Ladder (BioLabs); line 1–3, DNA fragment of 513 bp derived from mother plants tissues infected with F (1), FS (2), and CS (3); line 4–6, DNA fragment of 513 bp derived from first-generation seedlings tissues infected with F (4), FS (5), and CS (6); WC, water control. (**b**): membrane probed with CMV polyclonal antiserum (Bioreba AG). PC, positive control; 1–3, CP revealed from mother plants tissues infected with F (1), FS (2), and CS (3); 4–6, CP revealed from first-generation seedlings tissues infected with F (4), FS (5), and CS (6); HC, healthy control.

**Figure 4 plants-11-03217-f004:**
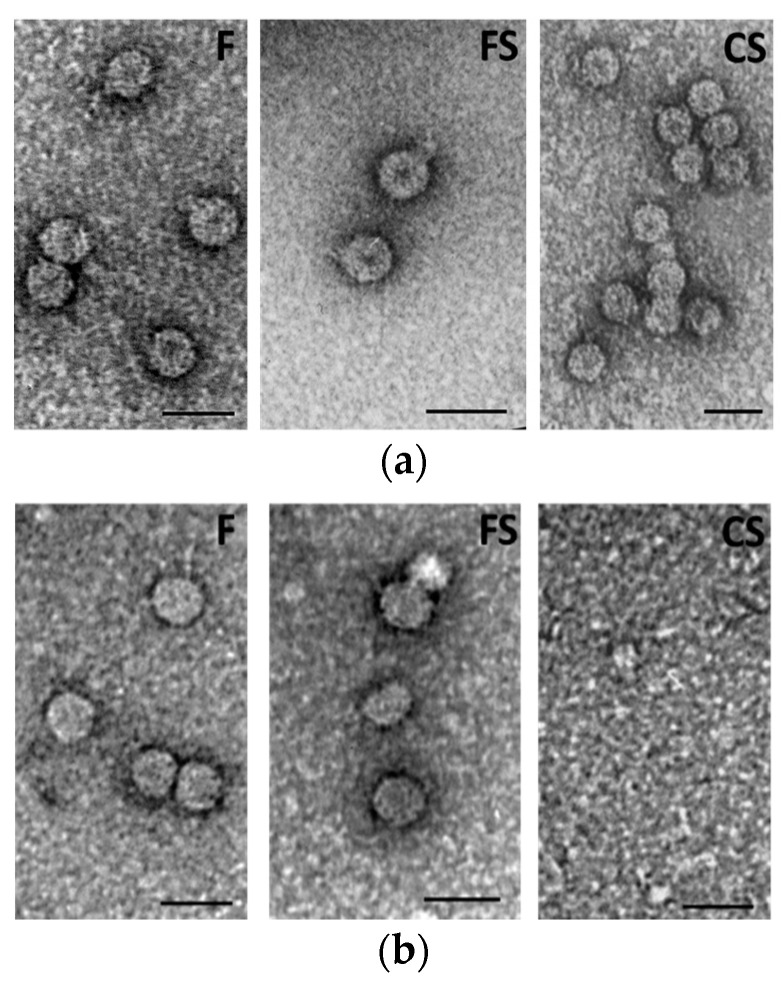
Electron micrograph of purified wild type (F), pseudorecombinant (FS) and chimeric (CS) virions isolated from tobacco mother plants (**a**) or from first-generation seedlings (**b**). Bar = 100 nm.

**Figure 5 plants-11-03217-f005:**
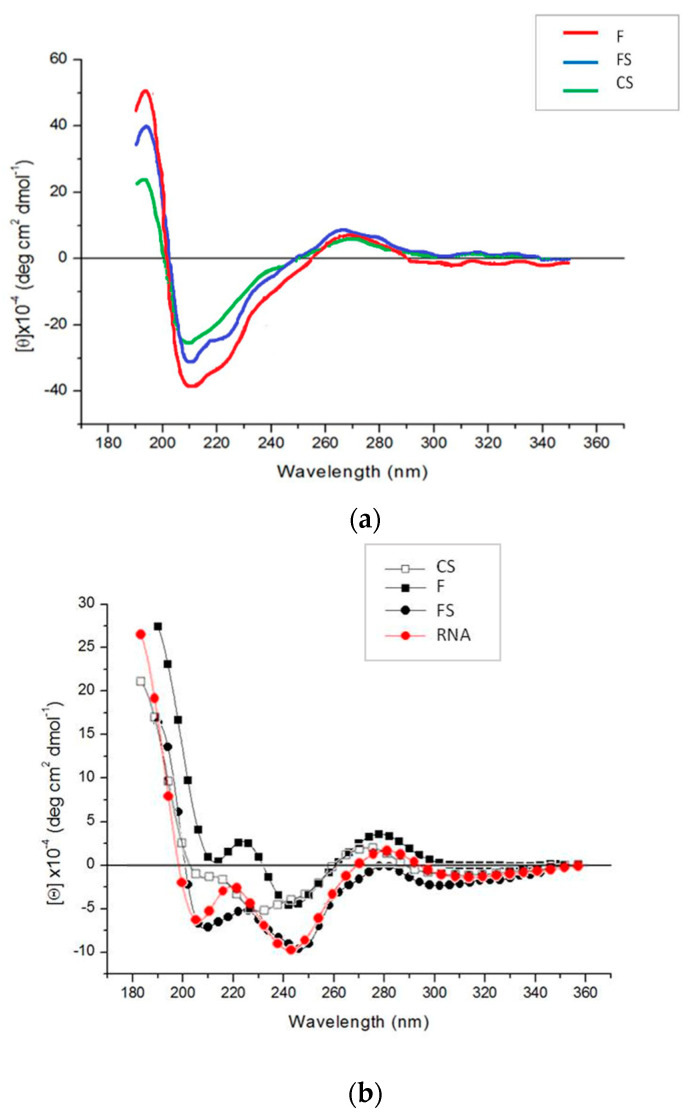
CD spectra of CMV purified wild type (F), pseudorecombinant (FS) and chimeric (CS) virions isolated from tobacco mother plants (**a**) or first-generation seedlings (**b**). CMV viral RNA was used to compare the CD spectra of first-generation seedling viruses. Note the different scale in each panel.

## Data Availability

Data presented in this study are available on request from the corresponding author.

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
