# Peer review of "Cucumber mosaic virus Is Unable to Self-Assemble in Tobacco Plants When Transmitted by Seed"

_plants, 2022, doi:10.3390/plants11233217_

Round 1

Reviewer 1 Report

Interesting study on a potential  mechanism controlling seed transmission of CMV and identification of a region of of the CMV coat protein controlling transmission and assembly of virions - linking the two processes for the first time. Although how this works mechanistically is still not clear, this is an important contribution to a neglected area of study.

Minor corrections

line 23 change to during the passage from one generation to the next

Line 30 change to Electron microscopy revealed absence (CS) of viral particles 

Line 210 change to... mother plants' viruses

but better to be more formal i.e. 'viruses extracted from mother plants ' 

Line 449 Section 5. 'Conclusions ' has been placed after the methods. Authors should check that they have entered the sections into the manuscript template in their intended order. This may not be rectifiable after manuscript publications.

line 452 The three different forms varied.. should be three different strains

line LINE  455: in this passgae re-specify the viral protein as being the CP otherwise you are confusing RNA and protein ..on the amino acids even- 454

tually implicated in this process and their position: codon for Ser131Gly131 (nt 391–393), 455 located in the βE-αEF region OF THE CP.

General presentation of figures is less than ideal - e.g. placement of panel labels and labeling of gel lanes looks like straight out of a student report and labeling should be made more 'journal-like'

Author Response

Dear Reviewer 1,

Thank you very much for accurate reading of the manuscript.

We have revised it, according to your comments.

We used the “Track Changes” function in order to highlight all corrections and any revisions, detailed below.

Interesting study on a potential mechanism controlling seed transmission of CMV and identification of a region of of the CMV coat protein controlling transmission and assembly of virions - linking the two processes for the first time. Although how this works mechanistically is still not clear, this is an important contribution to a neglected area of study.

Authors: thank you very much for your comment.

Minor corrections

line 23 change to during the passage from one generation to the next

Authors: the change has been done.

Line 30 change to Electron microscopy revealed absence (CS) of viral particles 

Authors: changed (lines 30-31).

Line 210 change to... mother plants' viruses but better to be more formal i.e. 'viruses extracted from mother plants ' 

Authors: we have changed it (new line 256).

Line 449 Section 5. 'Conclusions ' has been placed after the methods. Authors should check that they have entered the sections into the manuscript template in their intended order. This may not be rectifiable after manuscript publications.

Authors: we have used the journal manuscript template, where the “Conclusions” section is the number 5 and it follows “Materials and Methods” (section 4).

line 452 The three different forms varied. should be three different strains

Authors: we have replaced “forms” by “genotypes” (new line 510), here and throughout all the text.

line LINE  455: in this passage re-specify the viral protein as being the CP otherwise you are confusing RNA and protein on the amino acids eventually implicated in this process and their position: codon for Ser131→Gly131 (nt 391–393), located in the βE-αEF region OF THE CP.

Authors: thank you, the suggestion has been accepted (new line 514).

General presentation of figures is less than ideal - e.g. placement of panel labels and labeling of gel lanes looks like straight out of a student report and labeling should be made more 'journal-like'

Authors: thank you very much for this accurate control of figures. Figures 2, 3 (and their caption, accordingly), but also 1 and 5 have been modified in order to be clearer, better formatted, more formal and “journal-like”.

Author Response

There are no comments and suggestions by reviewer for authors to respond to. 

Reviewer 3 Report

The manuscript "Cucumber mosaic virus is unable to self-assemble in tobacco plants when transmitted by seed" describes a series of interesting experiments aimed at elucidating the transmissibility of Cucumber mosaic virus by seed in the Solanacaeus host plant - tobacco
The authors used plants artificially infected with purified virions of three genotypes of CMV virus - the CMV wild-type strain Fny, the pseudorecombinant CMV Fny/CMV-S, and a chimeric mutant CMV392mut. Seeds of these infected plants were harvested, and 400 randomly selected seeds were tested for virus infection. Seeds from the infected plants were retested for virus infection. Interestingly, there was no transmission from second generation seeds

I think such experiments are very interesting and important. While seed transmission is very important for virus spread and epidemiology, our knowledge of its mechanisms is rudimentary
However, I have some reservations about the completeness of the data presented by the authors
1. CMV is easily transmitted mechanically. It is not clear from the article whether the virus can be mechanically transmitted from both mother plants and the first generation of seed-infected plants. This is an important control that should be included
2. The authors present a number of interesting hypotheses that could explain the lack of transmission from the first generation of infected plants. These could be an interesting avenue for future research. One very likely hypothesis, in my opinion, is a shift in the sequence of the infecting CMV virus. The authors should definitely try to determine the sequence of the virus from the first generation of seed-infected plants. Or at least to check whether the sequence is identical to the original sequence
3. In relation to the above point, the authors have verified the presence of part of the RdRp gene using RT-PCR and the presence of CP using an immunoblot with CMV-specific commercial antibodies (see also the confusing wording in lines 145-146). No attempt was made to verify the presence of RNA2 and the integrity of individual RNAs
4. Figure 5A - the lines are too narrow, it is not easy to tell which line is which
5. The English language needs to be improved, some paragraphs are very difficult to read, e.g. lines 304-314, 402
-407.

I recommend acceptance of the manuscript after a major revision . 

Author Response

Dear Reviewer 3,

the authors are very grateful for the important comments and suggestions.

We have revised the manuscript, accordingly.

We used the "Track Changes" function in order to highlight all corrections and any revisions, detailed below.

I think such experiments are very interesting and important. While seed transmission is very important for virus spread and epidemiology, our knowledge of its mechanisms is rudimentary

Authors: the authors are grateful for this comment.

However, I have some reservations about the completeness of the data presented by the authors

  1. CMV is easily transmitted mechanically. It is not clear from the article whether the virus can be mechanically transmitted from both mother plants and the first generation of seed-infected plants. This is an important control that should be included

Authors: during our experiments, and in order to assess the integrity of viruses and their ability to infect healthy plants under horizontal mechanical transmission, we performed rutinary mechanical transmission tests of all three CMVs from mother plants to healthy tobacco plants (starting from both systemically infected leaves or purified virions). In all cases, the typical symptoms of CMV infection were observed in the inoculated plants at 12-14 d.p.i. Furthermore, amplicons of expected size confirmed the presence of the RdRp and CP genes (and also the exogenous fragment in CS) by RT-PCR and, finally, the CP was detected by Immuno-western blot and Immuno-dot blot analyses with CMV-specific antibodies.

In the first-generation seedlings, we did not perform these controls because we were interested to observe what happened only under strict vertical transmission condition. However, given our observations that viral particles were not correctly formed and that no further seed transmission was attained, it is very likely that CMV virions from the first-generation seedlings could not be mechanically transmitted.

For completeness, and in the light of your comment, we inserted the following sentences in the text:

 “As a control, virions from purified three CMV genotypes of mother plants were used to mechanically inoculate healthy plants. In all cases, RT-PCR analyses revealed the presence of the replicase and CP genes in all infected plant 14 d.p.i., also showed typical CMV symptoms and correct virions when observed under EM (data not shown).” (Results: new lines 112-116).

“Indeed, virions of the three CMV genotypes purified from mother plants were infectious and assembled correctly (see Results).” (Discussion: new lines 396-398).

  1. The authors present a number of interesting hypotheses that could explain the lack of transmission from the first generation of infected plants. These could be an interesting avenue for future research. One very likely hypothesis, in my opinion, is a shift in the sequence of the infecting CMV virus. The authors should definitely try to determine the sequence of the virus from the first generation of seed-infected plants. Or at least to check whether the sequence is identical to the original sequence
    3. In relation to the above point, the authors have verified the presence of part of the RdRp gene using RT-PCR and the presence of CP using an immunoblot with CMV-specific commercial antibodies (see also the confusing wording in lines 145-146). No attempt was made to verify the presence of RNA2 and the integrity of individual RNAs

Authors: thank you very much for your suggestion/hypothesis. During the present study, our attention was mainly captured by the possible role of RNA 3 (particularly CP). Certainly, we know that all three virus genotypes replicate and their CPs are traduced, as during our study we checked the presence of the CP by Immuno western blot in samples randomly chosen and derived from seedlings of first generation (also verifying the presence of the exogenous peptide by sequencing the CP). However, this does not preclude the accumulation of mutations in the virus genome, including RNA 3, that could explain why virions are not correctly formed and the virus cannot be seed transmitted. Unfortunately, we currently have no sequencing data to identify possible virus mutations that could involve RNA 1 or RNA 2 (or RNA 3) in first-generation seedlings. We agree with the reviewer that this is an exciting avenue of future research, which will allow verifying your hypothesis of eventual shift in the CMV sequence. We have added in the text some perspective of future possible studies aimed at more in-depth investigations in this regard (new lines 421-426).  

The sentence in lines 145-146 (new line 164-167) has been corrected to avoid confusion.

  1. Figure 5A - the lines are too narrow, it is not easy to tell which line is which

Authors: the figure 5a has been modified as suggested.

  1. The English language needs to be improved, some paragraphs are very difficult to read, e.g. lines 304-314, 402-407.

Authors: the authors are grateful for this comment. Some change has been performed throughout the text. In particular, the mentioned paragraphs have been thoroughly revised. We hope that now they are clearer and easier to read (new lines 333-367).

Round 2

Reviewer 3 Report

I thank the authors for adding important information to the manuscript. The authors have fully clarified my questions and addressed the uncertain points. I agree with the publication of the article in its present form.